# Is Social Participation Associated with Successful Aging among Older Canadians? Findings from the Canadian Longitudinal Study on Aging (CLSA)

**DOI:** 10.3390/ijerph20126058

**Published:** 2023-06-06

**Authors:** Mabel Ho, Eleanor Pullenayegum, Esme Fuller-Thomson

**Affiliations:** 1Factor-Inwentash Faculty of Social Work, University of Toronto, Toronto, ON M5S 1V4, Canada; 2Institute for Life Course and Aging, University of Toronto, Toronto, ON M5S 1V4, Canada; 3Child Health Evaluative Sciences, The Hospital for Sick Children Research Institute, Toronto, ON M5G 0A4, Canada; 4Dalla Lana School of Public Health, University of Toronto, Toronto, ON M5T 3M7, Canada

**Keywords:** Canadian Longitudinal Study on Aging (CLSA), older adults, social participation, sports, physical activities, exercises, volunteerism, successful aging

## Abstract

The present study examines various activities of social participation (i.e., church or religious activities; educational or cultural activities; service club or fraternal organization activities; neighbourhood, community, or professional association activities; volunteer or charity work; and recreational activities) as contributing factors to successful aging. Successful aging in this study includes the following: adequate social support, no limitations with respect to Activities of Daily Living (ADLs) and Instrumental Activities of Daily Living (IADLs), no mental illness in the preceding year, no serious cognitive decline or pain that prevents activity, as well as high levels of happiness, and self-reports of good physical health, mental health, and successful aging. **Methods:** The Canadian Longitudinal Study on Aging (CLSA) is a large, national, longitudinal study on aging. A secondary analysis of the baseline (i.e., 2011–2015) and Time 2 (i.e., 2015–2018) data of the CLSA was conducted on a sample of 7623 older adults who were defined as “aging successfully” at baseline and were aged 60+ at Time 2. Binary logistic regression analyses were employed to examine the association between engaging in various social participation activities at baseline and aging successfully at Time 2. **Results:** In a subsample (n = 7623) of the Canadian Longitudinal Study on Aging (CLSA) Comprehensive Cohort who were aging successfully at baseline, the prevalence of successful aging at Time 2 was significantly higher among the participants who participated in volunteer or charity work and recreational activities compared to those who were not involved in these activities. After adjusting for 22 potential factors, the results of the binary logistic regression analyses reported that participants who, at baseline, participated in volunteer or charity work and recreational activities had higher age–sex-adjusted odds of achieving successful aging (volunteer or charity work: aOR = 1.17, 95% CI: 1.04, 1.33; recreational activities: aOR = 1.15, 95% CI: 1.00, 1.32). **Conclusions:** Among six types of social participation activities, people who participated in volunteer or charity work and recreational activities were more likely to achieve successful aging than their counterparts who did not engage in these activities. If these associations are found to be causal, policies and interventions encouraging older adults to participate in volunteer or charity work and recreational activities may support older adults to achieve successful aging in later life.

## 1. Introduction

There is a paucity of studies that have examined successful aging in Canada, while most studies on successful aging have been conducted in the United States, Asia, and Europe (for important exceptions, please see [1,2]). Successful aging in this study includes the following: adequate social support, no limitations with respect to Activities of Daily Living (ADLs) and Instrumental Activities of Daily Living (IADLs), no mental illness in the preceding year, no serious cognitive decline or pain that prevents activity, high levels of happiness, and self-reports of good physical health, mental health, and successful aging [3].

Social participation is a critical determinant of successful aging [4]. Social activities often enhance social relationships and support and are significantly associated with individual health and well-being. It is estimated that 80% of older Canadians engage in at least one social activity (i.e., attending associations, church, clubs, educational activities, activities with family/friends, sports, volunteer or charity work, and others) [5]. Although most older Canadians are socially active, they are not physically active enough. Four in five (82.5%) of Canadian adults do not engage in sufficient physical activities (i.e., having at least 150 min of moderate- to vigorous-intensity physical activity per week as suggested by the Canadian Physical Activity Guidelines) [6,7], and are in sedentary behaviours most of the day (9.6 h per day). Only 4.5% of Canadians aged 60 and over have at least 30 min per day of physical activity, and 90% of them are in a sedentary state for at least 8 h per day [8].

## 2. Review of the Literature

### 2.1. Successful Aging

The successful aging construct used in the current study includes both objective and subjective measures of optimal aging [3]. From researcher-derived definitions, the construct includes the presence of physical, mental, and social functioning, psychological resources, and life satisfaction [9]. In addition, the new definition requires that neither memory problems, chronic pain that prevents activities nor mental illness was present in the past year. Unlike most earlier definitions of successful aging, those with chronic health conditions may still be classified as aging successfully as long as they are free of ADLs and IADLs and disabling pain. As suggested by Young et al.’s [10] (p. 87) multidimensional model of successful aging, successful aging “may coexist with disease and functional limitations if compensatory psychological and/or social mechanisms are used.” In this study, successful aging is defined as having no limitations with respect to Activities of Daily Living (ADLs) and Instrumental Activities of Daily Living (IADLs), no mental illness in the preceding year, no serious cognitive decline or pain that prevents activity, as well as having adequate social support, high levels of happiness, and self-reports of good physical health, mental health, and successful aging. The new construct has been widened to require that older adults report that they are positive about their own aging, physical health, mental health, and emotional well-being (e.g., happiness and/or life satisfaction).

### 2.2. Social Participation

Social participation is considered a critical determinant of successful aging because of its link to many health benefits, such as better cognitive health, health-related behaviors, self-rated health, and reduced risk of depression, disability, and mortality [5]. Social participation has been included in some theoretical models of successful and healthy aging (e.g., [4]). Social participation entails connecting with others, doing things with or for others, contributing to society, and/or receiving resources from society [11]. It includes social connections with families, friends, and communities, informal support to families, friends, and neighbours, community engagement, and volunteering. Social participation has shown positive effects on the well-being of older adults, such as improved health, health perception, quality of life, and life satisfaction [12,13,14,15]. It also contributes to social capital in which individual actions not only benefit the individuals but the whole community [16,17]. For example, a neighborhood watch group helps reduce the local crime rate and benefits all community members, including those who do not participate in the program. Some researchers have extended the definition of social participation to include visiting restaurants and bars, attending arts and cultural events, attending church, participating in other religious activities, and joining exercise groups and interest classes [18].

Using data from the 2008/2009 Canadian Community Health Survey (CCHS), Gilmour [5] examined the relationship between the frequency of participating in social activities and three outcomes relevant to well-being (i.e., positive self-perceived health, loneliness, and life dissatisfaction) among older Canadians. It was found that as the number of social activities increased, the strength between social participation and well-being outcomes increased. In addition, 21% of older men and 27% of older women wanted to engage in more social activities. Using the same dataset, Naud and his colleagues [19] compared social participation among older Canadian men and women by region and population size. They found that small cities had the highest social participation, while large cities had the lowest (17.4 vs. 14.3 social activities per month, respectively). They did not find any differences in social participation between men and women, but men (20.7%) and women (26.6%) wanted to participate more in social activities. They also found that men (27.1%) were more likely to report being “too busy” to participate in social activities than women (6.5%). In comparison, rural women (15.1%) were more likely to encounter transportation issues than rural men (1.2%) seeking to participate in social activities.

### 2.3. Conceptual Framework

The concept of “successful aging” has drawn international attention from scholars in a wide array of disciplines in the past 60 years [3]. It has generated a wealth of conceptual and empirical research over the last six decades, particularly in the last thirty years since Rowe and Kahn [20,21,22] proposed the MacArthur model of successful aging. The focus of the study of successful aging has shifted gerontology from “a discipline focused on disease and decline to one emphasizing health and growth” [23] (p. 201). In addition to researcher-defined classifications of successful aging, there are increasing empirical studies using lay perspectives in the literature on successful aging [24,25,26,27]. The theoretical underpinnings of this study draw upon earlier frameworks developed for use outside of gerontology: Bronfenbrenner’s hugely influential ecological systems theory [28,29], Keyes’s model of flourishing and complete mental health [30], and another integrated model of successful aging developed by Young and colleagues [10].

Guided by a conceptual framework synthesized by the three theoretical perspectives mentioned above, the concept of successful aging used in this study focuses on disability-free physical functioning, mental health, well-being, and social connectedness and includes the perspectives of the older adult. It is defined as having no limitations with respect to Activities of Daily Living (ADLs) and Instrumental Activities of Daily Living (IADLs), no mental illness in the preceding year, no serious cognitive decline or pain that prevents activity, and having adequate social support, high levels of happiness, and self-reports of good physical health, mental health, and successful aging. This expanded definition of successful aging considers both researcher-derived definitions and lay perspectives of successful aging.

This study aimed to examine the prevalence of successful aging among people who have engaged in six social participation activities, including church or religious activities; educational or cultural activities; service club or fraternal organization activities; neighbourhood, community, or professional services; volunteer or church work; and recreational activities. Implications for program and policy development and future research are discussed.

The present study examines the relationship between social participation and successful aging. Using baseline and Time 2 data from the Canadian Longitudinal Study on Aging (CLSA), the study aims to address the following research questions:

1a. Do those who participate in social activities at baseline have a higher prevalence of successful aging at Time 2?

1b. Do those who participate in social activities at baseline have higher age–sex-adjusted odds of successful aging at Time 2?

2. Does adjusting for 22 baseline characteristics attenuate the association between social participation and successful aging?

## 3. Methods

### 3.1. Study Population

This study analyzed data from the baseline (gathered 2011–2015) and Follow-up 1 (gathered in 2015–2018, hereafter referred to as Time 2 data) from the CLSA Comprehensive Cohort composed of 30,097 Canadian men and women. All participants aged 45 to 85 years at the time of recruitment are being followed for at least 20 years or until death [31]. Participants of the CLSA Comprehensive Cohort were interviewed at home. They underwent in-depth examination with the provision of biological specimens, such as blood and urine, at the CLSA Data Collection Sites. The CLSA has gone through 13 research ethics boards across Canada. Further information about the CLSA can be found at www.clsa-elcv.ca (accessed on 28 August 2022).

Of the 30,097 participants at baseline, 27,799 participants participated in Wave 2. Among them, 18,978 participants were 60 years or older at Time 2. The sample was restricted to those 60 and older who met the criteria of successful aging at baseline. Among these participants, 10,375 were excluded because they were not aging successfully at baseline, and 980 had missing entries in any analyzed variables at Wave 2. No individual variable had more than 5% missing. The final sample size was 7623 participants. This study involving secondary data analysis of CLSA data was approved by the Health Sciences Research Ethics Board of the University of Toronto (protocol number: 38284).

### 3.2. Measures

#### 3.2.1. Dependent Variable Assessed at Both Baseline and Time 2

**Physical Wellness**. Respondents were categorized as having physical wellness if they could perform all of their Activities of Daily Living (ADLs) and Instrumental Activities of Daily Living (IADLs) as listed below and did not have disabling pain and discomfort. ADLs included (1) dressing and undressing; (2) eating; (3) walking without help; (4) walking with some help including the use of a walking aid; and (5) getting in and out of bed. IADLs included (1) using the telephone; (2) getting to places out of walking distance; (3) shopping; (4) preparing meals; (5) doing housework; (6) doing housework with some help; (7) taking medicine; and (8) handling money.

**Psychological and Emotional Wellness**. Respondents were categorized as having psychological and emotional wellness if they did not have (1) depression [32,33]; (2) anxiety; (3) PTSD [34]; and they (4) felt depressed rarely or never, or some of the time; (5) felt happy occasionally or all of the time; (6) felt satisfied with life occasionally or all of the time [35]; and (7) did not have a memory problem.

**Social Wellness**. Respondents were categorized as having social wellness if they reported at least most of the time to have (1) someone to give them advice about a crisis; (2) someone to show them love and affection; and (3) someone to confide in or talk to about themselves or their problems.

**Self-Rated Wellness**. Respondents were categorized as having self-rated wellness if they reported that they perceived as “good” or “excellent” their own (1) aging; (2) physical health; and (3) mental health.

**Successful Aging.** Respondents were categorized as successful agers if they met all four criteria of physical, psychological, emotional, social, and self-rated wellness. Otherwise, they were categorized as typical agers. The primary focus of this study was to examine the relationship between social participation at baseline and successful aging at Time 2.

#### 3.2.2. Independent Variable Assessed at the Baseline Wave of Data Collection

Six social participation activities were examined. These variables were dichotomized from the five original response categories for three reasons: (1) some of the sample sizes of individual response categories were very small (i.e., <30); (2) the study was not intended to analyze the dose effect; and (3) there might be other reasons affecting the frequency of respondents’ participation in these activities.

**Church or religious activities.** Based on responses to the question that asked how often respondents engaged in church or religious activities such as services, committees, or choirs in the past 12 months (at least once a day, at least once a week, at least once a month, at least once a year, never), this variable was dichotomized as “never” versus “at least once in past year”.

**Educational or cultural activities.** Based on responses to the question that asked if respondents engaged in educational and cultural activities involving other people, such as attending courses, concerts, or plays or visiting museums in the past 12 months (at least once a day, at least once a week, at least once a month, at least once a year, never), this variable was dichotomized as “never” versus “at least once in past year”.

**Service club or fraternal organization activities.** Based on responses to the question that asked if respondents engaged in a service club or fraternal organization activities in the past 12 months (at least once a day, at least once a week, at least once a month, at least once a year, never), this variable was dichotomized as “never” versus “at least once in past year”.

**Neighbourhood, community, or professional association activities.** Based on responses to the question that asked if respondents engaged in neighbourhood, community, or professional association activities in the past 12 months (at least once a day, at least once a week, at least once a month, at least once a year, never), this variable was dichotomized as “never” versus “at least once in past year”.

**Volunteer or charity work.** Based on responses to the question that asked if respondents engaged in volunteer or charity work in the past 12 months (at least once a day, at least once a week, at least once a month, at least once a year, never), this variable was dichotomized as “never” versus “at least once in past year”.

**Recreational activities.** Based on responses to the question that asked if respondents engaged in any other recreational activities involving other people, including bridge, cards, hobbies, gardening, poker, and other games in the past 12 months (at least once a day, at least once a week, at least once a month, at least once a year, never), this variable was dichotomized as “never” versus “at least once in past year”.

#### 3.2.3. Covariates

In order to investigate what baseline factors were associated with successful aging at Time 2, we included in our analyses a wide range of socio-demographic characteristics as well as health behaviors and chronic health conditions. The exact questions used for these measures are available elsewhere (please see https://www.mdpi.com/1660-4601/19/20/13199# (accessed on 13 October 2022)).

### 3.3. Statistical Analysis

All analyses were conducted using SPSS Version 28. The percentages and odds ratios are weighted but the sample sizes are presented in their unweighted form. Bivariate analyses, including chi-square tests and *t*-tests, were conducted comparing successful agers to typical agers. Three binary logistic regression analyses were conducted with successful aging at Time 2 as the outcome and with the six social participation characteristics as the exposure of interest. In Model 1, only the social participation characteristics were included, and in Model 2, age and sex were added to the analysis. In Model 3, all the other baseline covariates were added to the model.

In order to ensure multicollinearity was not a problem, the variance inflation factor (VIF) and the Hosmer–Lemeshow test were used.

## 4. Results

### 4.1. Descriptive Statistics

The characteristics of the final sample (*n* = 7623, unweighted counts) and chi-square statistics (weighted percentages) are shown in Table 1. When asked “In terms of your own healthy aging, would you say it is excellent, very good, good, fair or poor?”, 97.5% of the respondents rated their own aging as good to excellent. In addition, more than seven in 10 (72.3%) of the respondents who rated their own aging as good to excellent were classified as successful agers using the expanded definition of successful aging presented in this study.

#### 4.1.1. Research Question 1a: Do Those Who Participate in Social Activities at Baseline Have a Higher Prevalence of Successful Aging at Time 2?

The results of the bivariate analyses indicate that the prevalence of successful aging at Time 2 was significantly higher in respondents who, at baseline, participated in educational or cultural activities (71.4% vs. 58.8%; *x*^2^(1) = 37.7, *p* < 0.001), those who participated in the neighbourhood, community, or professional association activities (72.2% vs. 67.5%; *x*^2^(1) = 18.7, *p* < 0.001), those who participated in volunteer or charity work (72.3% vs. 65.9%; *x*^2^(1) = 30.1, *p* < 0.001), and those who participated in recreational activities involving other people (71.3% vs. 66.3%; *x*^2^(1) = 12.6, *p* < 0.001). However, participation in church or religious activities and service club or fraternal organization activities were not significant in the bivariate analyses.

#### 4.1.2. Research Question 1b: Do Those Who Participate in Social Activities at Baseline Have Higher Age–Sex-Adjusted Odds of Successful Aging at Time 2?

In the fully adjusted model (see Figure 1), the odds of successful aging were significantly higher among older adults who, at baseline, participated in volunteer or charity work (aOR = 1.17, 95% CI: 1.04, 1.33) and recreational activities (aOR = 1.15, 95% CI: 1.00, 1.32) when compared to those who did not participate in these activities. However, participation in church or religious activities, educational or cultural activities, service club or fraternal organization activities, and neighbourhood, community, or professional association activities were not significant when age and sex were adjusted for.

#### 4.1.3. Research Question 2: Does Adjusting for 22 Baseline Characteristics Attenuate the Association between Social Participation and Successful Aging?

In the baseline model, the crude odds of achieving successful aging were about 60.1% more for older adults who participated in educational or cultural activities (OR = 1.60, 95% CI: 1.32, 1.94), 23.2% more for older adults who participated in volunteer or charity work (OR = 1.23, 95% CI: 1.09, 1.39), and 22.9% for older adults who participated in recreational activities (OR = 1.23, 95% CI: 1.08, 1.41) when only social participation activities were considered. In the full model, which adjusted for 22 factors, the odds of achieving successful aging for older adults who participated in volunteer or charity work (aOR = 1.17, 95% CI: 1.04, 1.33) and recreational activities (aOR = 1.15, 95% CI: 1.00, 1.32) were significant, but participation in educational or cultural activities was not statistically significant (aOR = 1.21, 95% CI: 0.99, 1.49, *p* = 0.068). Older adults who participated in these social participation activities (volunteer or charity work: 17.4–23.2%; recreational activities: 14.8–22.9%) were more likely to achieve successful aging than those not involved in these activities across all models.

### 4.2. Assessment of Model Fit

The results of the Omnibus Tests of Model Coefficients are highly significant (*x*^2^(38) = 378.1, *p* < 0.001), indicating that the final model is significantly better than the baseline model. Nagelkerke’s *R*^2^ equals 0.070, implying that the final model explains 7.0% of the variation in successful aging status. All variance inflation factors of the predictor variables ranged from 1.01 to 2.93 (VIF < 10), indicating that multicollinearity was not a concern.

## 5. Discussion

This study examined the relationship between social participation at baseline and subsequent successful aging. The findings indicate that the prevalence of successful aging was significantly higher among participants who participated in educational or cultural activities; neighbourhood, community, or professional association activities; volunteer or charity work; and recreational activities involving other people compared to participants who did not participate in these activities. However, when age and sex were taken into account, only engaging in volunteer or charity work and recreational activities at baseline was associated with Time 2 successful aging. After adjusting for 22 additional potential factors, the effect of engaging in volunteer or charity work and recreational activities was somewhat attenuated but remained significant.

Similar to the concept of successful aging [36], there is no agreement on social participation and how it can be measured [4,11]. Due to the lack of agreed-on standards for these two concepts, the findings of this study are difficult to compare with the findings of other studies directly. In addition, the construction of variables, such as successful aging and social participation, often relies on available variables in the dataset. Douglas and her colleagues [4] developed a model of social participation with three components: informal social participation, social connections, and volunteering. They found that social participation in all three domains was positively associated with physical and mental health. Their findings are consistent with other studies on the impact of social participation on physical and mental health [37,38]. This study found that participating in volunteer or charity work and recreational activities was positively associated with successful aging in later life. These findings are consistent with previous studies showing that participation in volunteer and/or recreational activities had positive effects on the well-being of older adults (i.e., improved health, health perception, quality of life, life satisfaction, psychological well-being, and social well-being) [12,13,14,15]. As little longitudinal research has been conducted to examine the relationship between social participation and successful aging, particularly taking into account many social activities, future research is required to replicate these findings and shed a greater understanding of the observed associations.

This study found that people who participated in volunteer or charity work had higher odds of achieving successful aging than those who did not. Research has shown that volunteering confers physical and mental health outcomes, but few studies have focused on how to encourage older adults to volunteer [39]. Warner and her colleagues [39] found that volunteering can be encouraged through face-to-face group-based interventions among older adults. Their randomized controlled trial indicated that older adults significantly increased their volunteering time after the intervention session. They suggested that “volunteering can be considered a strategy for successful aging because of its association with health, well-being, and longevity” [39] (p. 762). Volunteering has also been helpful in the clinical treatment of adolescent depression [40]. It may be useful to support older adults living with depression (e.g., mild depression) and tackle social isolation among older adults. Further research is required in these areas.

The present study also showed that participation in recreational activities was positively associated with successful aging. Research has shown that engaging in recreational activities or social activities in recreational settings is important to the well-being of older adults [14,18]. Participating in recreational activities can help tackle social isolation and loneliness in older adults by connecting them to the support they need and enhancing their experiences of belonging in their communities [14,41]. In addition to participation in volunteer activities, Ryu and Heo [13] also found that participation in social activities in recreational settings was positively associated with well-being in older adults. They suggested that participation in volunteering and recreational activities could improve the well-being of older adults and might contribute to successful aging. Encouraging and providing older adults with safe and appropriate opportunities for volunteering and recreational activities is particularly important following the impact of the COVID-19 pandemic [15].

Most Canadians are sedentary and do not engage in sufficient physical activities [6,7]. King [42] suggested ways to promote physical activity among older adults at the individual and societal levels. At the individual level, effective interventions included “behavioral or cognitive behavioral strategies” such as supporting older adults to set individual goals, engage in self-monitoring, and seek feedback and support from others [42] (p. 39). She also suggested promoting physical activity in the workplace and places of worship. At the societal level, adequate support involved the creation of “attractive, safe, and low-cost environments” such as local community recreation centres, community parks, walking and biking paths, and swimming pools [42] (p. 40). She also suggested creative uses of public places such as shopping malls and schools to encourage older adults to move and be active. If amenities for physical activity are available, accessible, and attractive to older adults, older adults may be more motivated to engage in an active lifestyle.

Recent studies have shown that using social prescribing to refer older adults to volunteering [43,44] and recreational activities [41,45] may be beneficial. Social prescribing, a way of integrating primary care and community services, can help improve patients’ health and well-being [46,47,48]. Social prescribing is defined as a process where healthcare professionals (social prescribers) can refer their patients to community agencies (social prescribing services) or agents (link workers) who can connect the patients with sources of support available in the community [49]. This process allows healthcare professionals and their patients to “co-design a non-clinical social prescription to improve their health and well-being” [49] (p.19). This shift from a clinical, pharmacological intervention to a social, non-pharmacological approach originated in the United Kingdom [47]. It has recently become popular in Canada and is supported by the College of Family Physicians of Canada as an effective way of using “social care for social needs” [48] (p. 88).

Understanding how an older person’s social context affects that person’s health and well-being is important. Under the biomedical model, healthcare professionals can manage their patients medically. However, they are often unprepared or supported to address their patients’ social needs and systemic barriers. This study found that engaging in volunteer or recreational work and recreational activities was positively associated with successful aging in later life. As many community organizations provide volunteering opportunities and recreational programs, healthcare providers can use social prescribing to support older adults in volunteer or charity work and recreational activities if appropriate.

### Limitations

The findings of this study should be interpreted in the context of the following limitations: First, as has been discussed elsewhere [3], due to the nature of secondary data analysis, we were constrained by the variables available within the CLSA data set. Unfortunately, salient concepts identified by Keyes [30] (p. 211) as relevant to complete mental health, such as acceptance of self and having a life purpose, were not available. Future research would benefit from investigation of other potentially important forms of social participation, such as social technology [50] or any “activities that provide interaction with others in society or the community” [11] (p. 2148). In addition, the observational nature of the study prohibited the determination of causality. For example, it may be that individuals who were volunteering and/or physically active at baseline differed fundamentally from their peers who were not involved in those activities. It may be that unaccounted-for variability that is associated with the successful aging at Time 2 rather than the activities themselves. To minimize this error, we restricted our sample to those aging successfully at baseline and controlled for a wide range of potential confounders. However, without a randomized control trial, it is impossible to know if the observed association is causal. Third, the participants were disproportionately well-educated with a post-secondary degree or diploma (79.5%). Therefore, the findings may be hard to generalize to the Canadian population, in which 55% of Canadians aged 65 years and over do not have a post-secondary education [51]. Fourth, although no individual variable had more than 5% missing, 980 were also excluded because they had missing entries in analyzed variables at Wave 2. Despite these limitations, the analyses of the baseline and Time 2 data of the CLSA provide valuable information on what social activities people participate in and the extent to which they are linked with successful aging and whether, taking into account a wide range of baseline factors, they attenuate the relationship between social participation and successful aging.

## 6. Conclusions

“Successful aging” is an important concept in gerontology [52]. Older adults worldwide want to achieve it. Policymakers want to develop policies to help older adults achieve it. Practitioners working with older adults want to provide interventions that support older adults in achieving it. Researchers in various fields want to determine what it is and develop an agreed-upon standard for measuring it. It has drawn the attention of gerontologists for over six decades [23,53]. In the current study, we have proposed a more inclusive definition of successful aging by using an expanded definition that includes both traditional elements from researcher-defined classifications, as well as lay definitions drawn from older adults themselves.

The present study found that people who participated in volunteer or charity work and recreational activities had higher adjusted odds of achieving successful aging than those who did not engage in these activities. It is suggested that social prescribing in volunteer opportunities and recreational programs may help support older adults’ health and well-being. Future research is needed to ascertain whether interventions in these areas result in increased odds of successful aging. Previous research has shown that Canadians are not physically active enough [6,7], and sedentary or inactive lifestyles have been shown to affect health [54]. Policies and interventions encouraging older adults to participate in volunteer or charity work and recreational activities, as well as engaging in an active and healthy lifestyle, may support older adults in achieving successful aging in later life.

## Figures and Tables

**Figure 1 ijerph-20-06058-f001:**
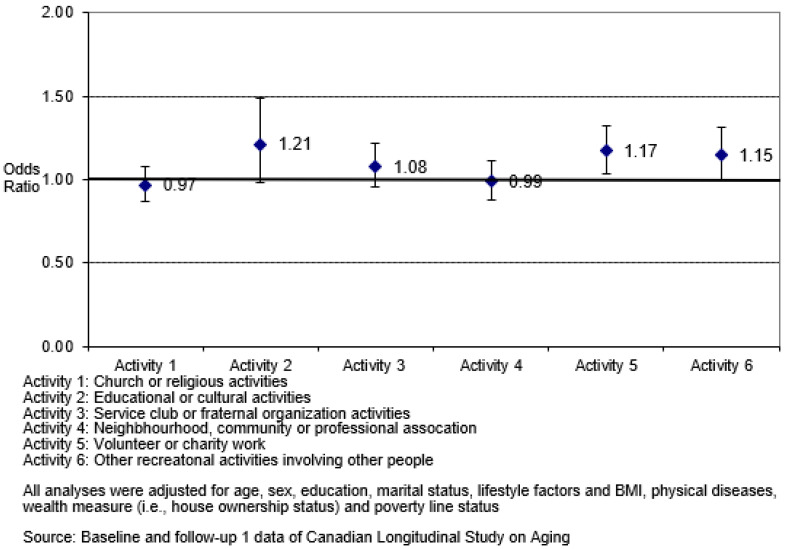
Adjusted odds ratio and 95% confidence interval of successful aging among participants who engaged in different social activities (*n* = 7623).

**Table 1 ijerph-20-06058-t001:** Description of the overall sample (*n* = 7623) with unweighted sample sizes and weighted percent.

Variables	Total		Successful Agers		Typical Agers		*x*^2^ (*df*) *p*-Value	% of Successful Agers
	Unweighted *n* = 7623	% Weighted	Unweighted *n* = 5373	% Weighted	Unweighted *n* = 2250	% Weighted		
**Church or religious activities**								
Never ever	3283	43%	2315	43%	968	43%	0.003 (1), *p* = 0.959	71%
Yes	4340	57%	3058	57%	1282	57%		71%
**Educational or cultural activities**								
Never ever	534	7%	314	6%	220	10%	37.7 (1), *p* < 0.001	59%
Yes	7089	93%	5059	94%	2030	90%		71%
**Service club or fraternal organization activities**								
Never ever	5352	70%	3763	70%	1589	71%	0.261 (1), *p* = 0.609	70%
Yes	2271	30%	1610	30%	661	29%		71%
**Neighbourhood, community, or professional association activities**								
Never ever	2805	37%	1894	35%	911	41%	18.7 (1), *p* < 0.001	68%
Yes	4818	63%	3479	65%	1339	60%		72%
**Volunteer or charity work**								
Never ever	2157	28%	1422	27%	735	33%	30.1 (1), *p* < 0.001	66%
Yes	5466	72%	3951	74%	1515	67%		72%
**Recreational activities**								
Never ever	1265	17%	839	16%	426	19%	12.6 (1), *p* < 0.001	66%
Yes	6358	83%	4534	84%	1824	81%		71%
**Sex**								
Male	3926	52%	2767	52%	1159	52%	0.00 (1), *p* = 0.992	71%
Female	3697	49%	2606	49%	1091	49%		71%
**Age groups (years)**								
55–59	1162	15%	878	16%	284	13%	188.8 (5), *p* < 0.001	76%
60–64	2123	28%	1625	30%	498	22%		77%
65–69	1701	22%	1229	23%	472	21%		72%
70–74	1115	15%	765	14%	350	16%		69%
75–79	1015	13%	611	11%	404	18%		60%
80+	507	7%	265	5%	242	11%		52%
**Education**								
<Secondary school graduation	311	4%	185	3%	126	6%	24.1 (2), *p* < 0.001	60%
Secondary school graduate and/or with some post-secondary education	1221	16%	833	16%	388	17%		68%
Post-secondary degree/diploma	6091	80%	4355	81%	1736	77%		72%
**House ownership**								
Paying rent	944	12%	582	11%	362	16%	41.1 (2), *p* < 0.001	62%
Paying mortgage	1634	21%	1159	22%	475	21%		71%
Paid off mortgage	5045	66%	3632	68%	1413	63%		72%
**Poverty line status**								
Under poverty line income	172	2%	88	2%	84	4%	97.6 (3), *p* < 0.001	51%
Marginal income	1476	19%	928	17%	548	24%		63%
Above poverty line income	5492	72%	4033	75%	1459	65%		74%
No answer	483	6%	324	6%	159	7%		67%
**Marital status at baseline**								
Single	355	5%	218	4%	137	6%	68.1 (3), *p* < 0.001	61%
Married	5862	77%	4262	79%	1600	71%		73%
Widowed	685	9%	412	8%	273	12%		60%
Divorced/separated	721	9%	481	9%	240	11%		67%
**BMI**								
Underweight/normal weight	2376	31%	1737	32%	639	28%	25.4 (2), *p* < 0.001	73%
Overweight	3375	44%	2398	45%	977	43%		71%
Obese	1872	25%	1238	23%	634	28%		66%
**Smoking status**								
Never smoked	2504	33%	1802	34%	702	31%	10.7 (2), *p* < 0.01	72%
Former smoker	4821	63%	3383	63%	1438	64%		70%
Current smoker	298	4%	188	4%	110	5%		63%
**Sitting activity**								
Never/seldom	114	2%	75	1%	39	2%	1.23 (1), *p* = 0.268	66%
Sometimes/often	7509	99%	5298	99%	2211	98%		71%
**Walking**								
Never/seldom	1931	25%	1296	24%	635	28%	14.1 (1), *p* < 0.001	67%
Sometimes/often	5692	75%	4077	76%	1615	72%		72%
**Light sports**								
Never/seldom	6721	88%	4708	88%	2013	90%	5.17 (1), *p* = 0.023	70%
Sometimes/often	902	12%	665	12%	237	11%		74%
**Moderate sports**								
Never/seldom	7096	93%	4962	92%	2134	95%	15.3 (1), *p* < 0.001	70%
Sometimes/often	527	7%	411	8%	116	5%		78%
**Strenuous sports**								
Never/seldom	5928	78%	4079	76%	1849	82%	36.0 (1), *p* < 0.001	69%
Sometimes/often	1695	22%	1294	24%	401	18%		76%
**Muscle and endurance exercises**								
Never/seldom	6007	79%	4212	78%	1795	80%	1.82 (1), *p* = 0.177	70%
Sometimes/often	1616	21%	1161	22%	455	20%		72%
**Sleep problem**								
Never/rarely/some of the time	5655	74%	4073	76%	1582	70%	25.0 (1), *p* < 0.001	72%
Occasional/all of the time	1968	26%	1300	24%	668	30%		66%
**Diabetes**								
No	6444	85%	4587	85%	1857	83%	9.77 (1), *p* < 0.005	71%
Yes	1179	16%	786	15%	393	18%		67%
**Heart disease**								
No	6750	89%	4831	90%	1919	85%	33.4 (1), *p* < 0.001	72%
Yes	873	12%	542	10%	331	15%		62%
**Hypertension**								
No	4702	62%	3405	63%	1297	58%	22.0 (1), *p* < 0.001	72%
Yes	2921	38%	1968	37%	953	42%		67%
**Arthritis**								
No	6923	91%	4897	91%	2026	90%	2.29 (1), *p* = 0.131	71%
Yes	700	9%	476	9%	224	10%		68%
**Osteoporosis**								
No	6860	90%	4859	90%	2001	89%	3.96 (1), *p* = 0.047	71%
Yes	763	10%	514	10%	249	11%		67%

## Data Availability

The data are available from the Canadian Longitudinal Study on Aging (www.clsa-elcv.ca (accessed on 28 August 2022)) for researchers who meet the criteria for access to de-identified CLSA data.

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
