# Peer review of "Is Social Participation Associated with Successful Aging among Older Canadians? Findings from the Canadian Longitudinal Study on Aging (CLSA)"

_ijerph, 2023, doi:10.3390/ijerph20126058_

Round 1

Reviewer 1 Report

This is a novel and compelling manuscript that explores the impact of social participation on successful aging. I have some suggestions as follows:

1. Although there is no agreement on social participation and how it can be measured, many works reveal its relation to health. I suggested more explanation was needed. Such as why volunteer or charity work and recreational activities out of the six types of social participation activities are related to successful aging.

2. Why the author mentioned “social prescribing”, and “It is suggested that social prescribing in volunteer opportunities and recreational programs can help support older adults’ health and wellbeing.” Why is “social prescribing” related to volunteer or charity work and recreational activities?

3. Does the information about the missing data need to be listed?

Author Response

Thank you for your time and comment. Please see the attachment.

Reviewer 2 Report

Thanks for submitting the manuscript. I think there are several major issues need to be addressed by the authors:

1) The authors need to provide more discussion on how to conceptualize “successful aging”.

2) Please also provide more detail operationalization of the “successful aging”. This is a very important variable in this study.

3) My major concern is the sample size, there were 30,097 participants in total, but why the authors only selected 7,623 samples. Please provide more detail justification, including the inclusive and exclusive criteria.

4) Please provide the justifications for dichotomize the DV (p. 4-5-). Combining at least once a day and once a year in a group is a bit problematic. I think the authors should keep the ordinal measurement.

5) There are also concerns about the data analysis methods, as the data is longitudinal in nature it is better to use longitudinal data analysis methods, such as latent growth curve model, etc.

6) There hypothesis and theoretical framework are not clear. Please strength it.

7) Being volunteer and the respondents’ income are closely related. The authors may also consider to explore the mediating effects between the variables.

Author Response

(The authors gave the same response as above.)

Reviewer 3 Report

The authors used data from Canada to explore the relationship between social participation and successful aging. The topic is interesting and the findings are interesting. While the paper is well-presented, there are a few areas that need to be improved before publication.

1. The literature review is not enough. The authors described the findings from other studies but did not dive into how social participation could affect successful aging. For example, if looking through Coleman's social capital theory, social participation could allow good habits to diffuse thus improving participants' health, and vice versa. The significance of the content would be much compromised if the author do not try to probe the mechanisms. I believe if the authors do, there could be a much more interesting discussion on why participation in certain activities is conducive to successful aging.

2. The authors should avoid using "as discussed elsewhere" to refer to their previous publication (Ho et al, 2022). Instead, the authors should try to mention what core arguments they are referring to.

3. The part of the discussion is irrelevant. The authors talked a lot about social prescribing and sleeping while the empirical evidence really did not tell much about them. Would it be better if the authors move these to the literature review? Now the discussion is distracting, and the authors should make efforts to stay focused on the empirical results, and then expand on explaining the results. 

4. The contribution is not enough. Social participation and health, or healthy aging is not a new topic, at all. The contribution of the research cannot simply be grounded on the paucity of research in one region, in this case, Canada. The authors should think more about and highlight the contribution/significance. 

Author Response

(The authors gave the same response as above.)

Round 2

Reviewer 2 Report

Thanks for submitting the revised manuscript. I am fully satisfied with the  responses and changes made by the authors.

Reviewer 3 Report

The authors have addressed my concerns. I believe this manuscript is in a publish-ready form.